# Phenotypic Plasticity and Local Adaptation of Leaf Cuticular Waxes Favor Perennial Alpine Herbs under Climate Change

**DOI:** 10.3390/plants11010120

**Published:** 2021-12-31

**Authors:** Luhua Yao, Dengke Wang, Dangjun Wang, Shixiong Li, Youjun Chen, Yanjun Guo

**Affiliations:** 1College of Grassland Science, Qingdao Agricultural University, Qingdao 266109, China; ylh20210625@126.com; 2College of Agronomy and Biotechnology, Southwest University, Chongqing 400716, China; wdk_1314@126.com (D.W.); twwangdj@163.com (D.W.); 3Qinghai Academy of Animal Science and Veterinary Medicine, Xining 810016, China; shixionglee@hotmail.com; 4Institute of Qinghai Tibetan Plateau, Southwest Minzu University, Chengdu 610041, China; chenyoujun2005@163.com

**Keywords:** adaptation, alpine meadow plants, climate change, cuticular waxes, phenotypic plasticity

## Abstract

Six perennial herbs (*Plantago asiatica*, *Polygonum viviparum*, *Anaphalis lactea*, *Kobresia humilis*, *Leontopodium nanum* and *Potentilla chinensis*) widely distributed in alpine meadows were reciprocally transplanted at two sites in eastern edge of Qinghai-Tibetan Plateau, Hongyuan (3434 m, 2.97 °C, 911 mm) and Qilian (3701 m, 2.52 °C, 472 mm), aiming to evaluate the responses of alpine plants to changing environments. When plants were transplanted from Hongyuan to Qilian, most plant species showed a decrease of total wax coverage in first year and reverse trend was observed for some plant species in second year. However, when plants were transplanted from Qilian to Hongyuan, the response of total wax coverage differed greatly between plant species. When compared with those in first year, plasticity index of average chain length of alkane decreased whereas carbon preference index of alkane increased at both Hongyuan and Qilian in second year. The total wax coverage differed between local and transplanted plants, suggesting both environmental and genetic factors controlled the wax depositions. Structural equation modeling indicated that co-variations existed between leaf cuticular waxes and leaf functional traits. These results suggest that alpine herbs adjust both wax depositions and chain length distributions to adapt to changing environment, showing climate adaptations.

## 1. Introduction

Alpine grasslands are currently facing great challenges from global climate change, which alters the surroundings suitable for plants. Such changes in turn let the plants adjust themselves to actual or expected climate, showing climate adaptation [1]. For an individual plant, its ability of adaptation can be influenced by both intraspecific variation (genotype or genetic difference) and the environment [2,3]. The phenotypic plasticity is the ability of individual to develop different phenotypes in various environments, resulting into intraspecific variations [2,4]. Cuticular waxes cover the surface of land plants and play pivotal roles in protecting plants from abiotic stresses. However, the phenotypic plasticity of cuticular waxes of alpine plants under climate change is still unclear.

Cuticular waxes, as the outmost surface of plant, are mixtures of dozens of hydrophobic compounds, mostly very-long-chain fatty acids and their derivatizes such as aldehydes, primary alcohols, alkanes, alkyl esters, ketones, secondary alcohols, and triterpenoids [5]. Studies have shown that the cuticular waxes are very sensitive to changing environments, with their crystalloid morphology, wax compositions, or deposition showing adaptation alteration according to surroundings [6]. For example, an increase of total wax amount, particularly alkanes, is regarded as plant common response to drought stress for most plant species. The wax crystalloid will melt under enhanced UVB irradiation, enlarging wax coverage over leaf surface which prevents the over dose of UVB entering the epidermis. In alpine meadow ecosystems, the total wax amount and wax composition of alpine meadow plants were altered in responding to changing annual temperature and aridity index [7]. Bush and McInerney [8] also reported that mean annual temperature was most significant for synthesis of the longer chain hydrocarbons. Relative to humid and cold environments, the plants in dry and high temperature environments had obvious predominance of long chain alkane [9]. Dodd and Poveda [10] also reported that greater weighted mean alkane chain length at low and at high elevations was possibly a result of adaptation to minimize cuticular permeability due to high summer temperatures at low elevation and freezing causing physiological drought at high elevations. Therefore, the phenotypic plasticity of cuticular waxes is essential for plants adapting to the new environments induced by climate change. Meanwhile, the alterations of leaf cuticular wax may influence forage digestibility by animals [11], oviposition of insect on plant leaves [12], and attachment and effectiveness of predatory insects [13]. Therefore, Understanding the phenotypic plasticity of cuticular waxes can further explain why plants can adapt to changing environmental conditions during their life cycle and their potential influences on ecological systems [14,15].

Besides cuticular waxes, leaf functional traits such as leaf size, width, thickness, specific leaf area (SLA), and chlorophyll content, etc, are also sensitive to changing environments. In a rapidly changing climate, quick responses of leaf functional traits are important for plants to persist within communities and to better tolerate changing environmental conditions [16]. For example, narrow leaves are associated with higher mean annual precipitation in Protea [17]. A study in a Mongolian steppe has shown that leaf traits plasticity played a critical role in vulnerability of species to a rapidly changing environments [18]. Leaf morphology differences between ecotypes of *Senecio lautus* have been shown to be a result of divergent selection on leaf shape or associated traits that confer an adaptive advantage in each environment [19]. However, the adaptation of a plant in changing environments might be the result of performance trade-offs induced by different leaf functional traits. For example, to fuel growth, light-demanding species in the moist forest had a high leaf area ratio, whereas, to reduce transpiration and water loss, light-demanding species in the dry forest had a low leaf area ratio [20]. Knowledge of the phenotypic plasticity of both cuticular waxes and other leaf functional traits will benefit to our understanding of plant responses to changing environments.

Reciprocal transplanting is an important experimental method to elucidate the contributions of environments and genetic factors on plant phenotypic plasticity [21]. A study in east of Qinghai-Tibet Plateau has shown that transplanting the *Albies faxoniana* from 3500 m to 2850 m reduced the leaf chlorophyll content and water utilization efficiency and increased the photosynthesis [22]. In Himalayas, transplanting *Viola biflora* var. rockiana to warmer sites limited population growth rates, survival and clonality, and increased the plant community productivity [23]. A reciprocal transplanting study of *Primula nutans* ssp. finmarchica has also shown that the plants were already suffering from adaptational lag due to climate change, and further warming might increase this maladaptation [24]. This suggests that reciprocal transplanting not only can explain plant phenotypic plasticity but also can elucidate the principal environmental factors contributing to plant adaptation. The broad niche of plant species under various environments is due to its ability to express adaptive phenotypic plasticity, not to locally adapted genotypes. For example, environmental variation across alpine regions, potentially the contrasting precipitation pattern, is a strong driver of local adaptation [25]. Factors affecting the leaf wax deposition are various, such as soil water condition, air temperature and humidity, precipitation, shade, and UVB irradiation, etc. [15]. The wax characteristics in a transplanting environment might be an adaptation to comprehensive environmental conditions, resulting from trade-offs between responses to each individual factor. Therefore, we hypothesized that intraspecific leaf wax trait variations might be the results of quick adaptation of plants to changing environments.

To investigate the phenotypic plasticity of cuticular waxes of alpine meadow plants as well as leaf functional traits under climate change, we conducted a reciprocal transplant experiment on the Qinghai-Tibetan Plateau over three years (2017–2019). Six perennial herbs that are widely distributed in this area were reciprocally transplanted at two sites, N 32° E 102° (Hongyuan) and N 37° E 100° (Qilian). We then tested the variations of leaf cuticular waxes as well as other leaf functional traits such as leaf size, width, thickness, specific leaf area (SLA), and chlorophyll content, for the following consecutive years. In a latest research, Guo et al. [26] reported that long-term adaptation under certain environments would induce genetic mutation of wax biosynthesis genes, resulting in inheritable alterations of cuticular wax depositions. Therefore, we hypothesized that phenotypic plasticity of plant cuticular waxes of alpine meadow will benefit plant adaptability under changing environments.

## 2. Results

The leaf traits of alpine meadow plants showed great variations between years, plant species, experimental sites and transplanting (Table 1), suggesting that plant would adjust its functional traits to adapt to the changing climate conditions. Here, we mainly explored how the leaf cuticular waxes of alpine meadow plants responded to the quick environment changes induced by reciprocal transplanting. 

### 2.1. Responses of Leaf Wax Coverage and Wax Compositions

The total wax coverage of alpine plants was significantly influenced by plant species, sites, years, transplanting, and their interactions (Table 1 & Figure 1). At Hongyuan site, total wax coverage on leaves of *P. asiatica* showed no difference between local and transplanted plants in both years (Figure 1a). The total wax coverage on leaves of local *A. lactea* and *P. viviparum* were significantly higher than those of transplanted plants in 2018 but showed no difference in 2019 (Figure 1b,c). Total wax coverage of local was higher than those of transplanted plants for *L. nanum* but was lower than those of transplanted plants for *P. chinensis* in 2019 (Figure 1e,f). At Qilian site, the total wax coverage of local plants was significantly lower than those of transplanted plants for *P. asiatica* in both 2018 and 2019, and for *A. lactea* and *P. chinensis* in 2019 (no difference in 2018) (Figure 1a,c,f). However, the total wax coverage of local plants was higher than those of transplanted plants for *P. viviparum* and *L. nanum* in 2018 and for *K. humilis* in 2019 (Figure 1b,d,e). Year variation of total wax coverage was also observed for all plant species. Overall, at Hongyuan site, when compared to 2018, total wax coverage of transplanted *P. viviparum* and *A. lactea* reduced by 51.32% and 98.07% in 2019, respectively, that of *P. chinensis* increased by 2.439 times, whereas that of *P. asiatica* and *L. nanum* changed insignificantly. At Qilian site, when compared to 2018, total wax coverage of transplanted plants increased in 2019. 

The main wax compositions of the six tested plant species included *n*-alkane, *n*-alcohol, *n*-alkanoic acid and aldehyde (Figure 2). However, their wax profiles differed greatly between the six plant species. For example, diketone was observed only in *P. viviparum* and triterpenoids (amyrin) were observed only in *P. chinensis* (Figure 2b,f). Transplanting significantly influenced the wax profiles. Alkane, as the predominant wax component of *P. asiatica* and *A. lactea*, its amount was significantly higher in transplanted plant than in local plant at Qilian, whereas opposite trend was observed at Hongyuan site (Figure 2a,c).

### 2.2. The Response Coefficient (RC) and Phenotypic Plasticity Index (PI) of Leaf Wax Traits 

The response coefficient (RC) of wax traits varied from 2018 to 2019 at Hongyuan and Qilian (Table 2). Overall, the HT/HL HT/QL and QT/QL of wax components were larger than 1, showing that *P. asiatica* planted in Hongyuan had higher wax coverage. For *A. lactea*, the content of wax increased at Qilian than Hongyuan in 2019. For *P. viviparum* and *P. chinensis*, transplanting speies at Qilian had higer wax coverage in 2019. The wax coverage of transplanted plants had reduced trend for *L. nanum* and *K. humilis* at both Hongyuan and Qilain site, showed that new environment was unfavorable to wax deposition.

The cuticular wax showed different phenotypic plasticity at two environmental sites (Figure 3 & Appendix A). We found that the phenotypic plasticity of alkane and total wax varied significantly between Hongyuan and Qilian and between 2018 and 2019 (Figure 3a,b). Overall, for plants transplanted to Hongyuan, the phenotypic plasticity of alkane and total wax varied from 0.297 (*P. asiatica*) to 0.686 (*A. lactea*) and 0.147 (*P. viviparum*) to 0.366 (*P. chinensis*) in 2018, and from 0.128 (*P. asiatica*) to 0.935 (*A. lactea*) and from 0.147 (*P. asiatica*) to 0.926 (*A. lactea*) in 2019, respectively. When *K. humilis* was transplanted to Hongyuan from Qilian, no survival was observed. For plants transplanted to Qilian, the phenotypic plasticity of alkane and total wax indices varied from 0.078 (*P. asiatica*) to 0.807 (*P. chinensis*) and 0.185 (*P. asiatica*) to 0.758 (*A. lactea*) in 2018, and from 0.175 (*P. asiatica*) to 0.981 (*A. lactea*) and from 0.174 (*P. asiatica*) to 0.974 (*A. lactea*) in 2019, respectively (Figure 3c,d). Interestingly, among the plants transplanted from Qilian to Hongyuan, *P. chinensis* relatively had higher plasticity of both wax and leaf morphology traits. whereas the plants transplanted from Hongyuan to Qilian, *A. lactea* and *L. nanum* had higher plasticity of both wax traits and leaf morphology traits. 

The phenotypic plasticity index of the chain length distributions of main wax composition was further calculated (Figure 3). For example, ACL_alkane_ of plants in Hongyuan varied from 0.002 (*P. viviparum*) to 0.062 (*P. chinensis*) in 2018 and from 0.005 (*P. asiatica* and *L. nanum*) to 0.017 (*P. viviparum*) in 2019, respectively; whereas CPI_alkane_ varied from 0.033 (*P. asiatica*) to 0.826 (*P. viviparum*) in 2018 and from 0.057 (*P. chinensis*) to 0.222 (*P. asiatica*) in 2019, respectively (Figure 3a,b). For plants in Qilian, ACL_alkane_ varied from 0.002 (*P. viviparum*) to 0.055 (*P. chinensis*) in 2018 and from 0.001 (*P. chinensis*) to 0.028 (*P. asiatica*) in 2019, respectively; whereas CPI_alkane_ varied from 0.019 (*A. lactea*) to 0.915 (*P. viviparum*) in 2018 and from 0.027 (*P. chinensis*) to 0.900 (*L. nanum*) in 2019, respectively (Figure 3c,d). Overall, when compared with those in 2018, the PI of ACL_alkanes_ decreased whereas CPI_alkanes_ increased in both Hongyuan and Qilian. Among plant species, the PI of ACL_alkanes_ increased for *P. asiatica* and *P.viviparum* but reduced for *A. lactea* and *P. chinensis*, whereas CPI_alkanes_ increased for *P. asiatica* and *L. nanum* but reduced for *P. viviparum* and *P. chinensis* in Hongyuan and Qilian. However, PI of ACL_alcohols_ decreased for *P. chinensis* by 57% in Hongyuan while increased by a factor of 1 in Qilian from 2018 to 2019. For *K. humilis*, both PI of ACL_alkanes_ and ACL_alcohols_ decreased in Qilian.

### 2.3. Covariation of Phenotypic Plasticity Indices among Traits

According to the result of principal component analysis (PCA), the first two principal components (PCs) together described ~73% of the total variation in all measured functional traits across plant species at Hongyuan and Qilian (Appendix A). The PCA revealed several gradients of among-species trait covariation (Figure 4). At Hongyuan site, the first PCA axis reflected PI of leaf morphological traits, the second axis reflected mostly variations in wax traits (Appendix A). The first principal component (PC1) had high positive loadings for leaf morphological traits, whereas PC2 had high negative loadings for leaf wax traits and leaf thickness (Appendix A). At Qilian, the first PCA axis reflected PI of chlorophyll and wax traits, the second axis reflected mostly variation in height and leaf thickness. The first principal component (PC1) had high negative loadings for height, chlorophyll and leaf morphological traits (except leaf specific area) and high positive loadings for leaf wax traits and leaf specific area. Overall, the wax traits explained 38.48% of total variation at Hongyuan, whereas wax traits and leaf morphological traits had covariation and together explained 72.52% total variation at Qilian. 

The structural equation model (SEM) analysis showed that, when environmental resources changed from “better” (Hongyuan) to “bad” (Qilian), the plant wax coverage and alkane content directly affected the wax plasticity, whereas leaf traits indirectly affected wax plasticity by altering the alkane content (Figure 5a and Appendix A). When environmental resources changed from “bad” (Qilian) to “better” (Hongyuan), the leaf traits, wax coverage and wax composition content had no effect on wax plasticity, whereas leaf traits directly affected the wax coverage and wax composition content (Figure 5b). 

## 3. Discussion

### 3.1. Cuticular Waxes of Plants Were Changed in Order to Adapt to New Environments

Cuticular waxes are the plant’s outermost barrier between the plant and its environment, and previous studies showed that the plant’s cuticle structure and chemical compositions would change substantially in response to environmental stresses (such as temperature, water, pathogen, and phytophagous insect) [27,28]. A number of studies showed that the climatic stressors such as increased temperatures and frequent drought periods were the main effects leading to heavier wax loads, changes and accumulation in the chemical composition of waxes [14,29,30]. A study in Qinghai-Tibetan Plateau also showed that the averaged amounts of wax compositions and total cuticular waxes of alpine meadow plants were significantly correlated with the mean annual temperature and aridity index [7]. In this study, the plant species differed in their responses to environmental change. When facing same environmental change, total wax coverage increased in some plant species but reduced or unchanged in other species (Figure 2 & Appendix A). These results similarly showed that the change of cuticular waxes were mainly induced by climate changes such as annual precipitation and temperature. More importantly, such difference might contribute to the alterations of plant communities under long term climate change [31]. 

Additionally, transplanting created a sharp change of environments especially precipitation, which significantly resulted in alterations of leaf wax depositions on alpine meadow plants, suggesting that the alterations of leaf waxes might be a common response of alpine plant to adapt to environmental changes. A study in the Swiss Alps also has shown that land use and water availability drive community-level plant functional diversity of grasslands along a temperature gradient, attributing to plant different acquisitive strategies [32]. However, soil properties showed no interaction with temperature to affect plant trait variations [32]. Therefore, the difference in climatic factors but not soil properties at two experimental sites might be the main reason driving the alterations of leaf wax depositions [33,34]. Overall, when plants were transplanted from Hongyuan (“better condition”) to Qilian (“bad condition’), most plant species showed a decrease of total wax coverage when compared to their original compartments one year after transplanting (in 2018). However, when plants were transplanted from Qilian to Hongyuan, the response of total wax coverage differed between plant species (Figure 2). This further clarified that the alterations of wax depositions differed under different environmental pressures. In 2019, two years after transplanting, total wax coverage of some plant species showed opposite trend when compared with those in 2018. This might be correlated to the climate adaptation of the transplanted plants [35], which suffered from “new” environment at the beginning and acclimated their physiological characteristics to new habitats [36]. The plants were transplanted with the sod and recovered for one year before sampling, therefore, these regrew perennial plants already acclimated to the new environments and their responses might be physiological adaptations to climate change. However, handling effects (transplanting) might also influence plant responses to new environment, which might be minor but still need consideration in the future studies. 

To further demonstrated the environmental adaptations of leaf cuticular waxes, the phenotypic plasticity index was calculated for wax trait as well as other leaf functional traits (Figure 4 & Appendix A). Overall, the cuticular wax showed different PI at two environmental sites. The plants transplanted from Qilian to Hongyuan, showed similar leaf wax plasticity for *P. asiatica*, *P. viviparum*, *A. lactea* and *L. nanum*, whereas for plants transplanted from Hongyuan to Qilian, the leaf wax traits of *P. asiatica*, *P. viviparum* and *A. lactea* had higher plasticity, whereas the plasticity of leaf morphology traits decreased, particularly plant height and leaf length. These results indicated that plants might balance between their functional traits to adapt to the changing environmen [37], and trade-off might exist among leaf functional traits [38]. The chain length distribution of plant long chain alkanes is an important index indicating climate history. For example, when compared with high latitudes, plant from low latitudes had higher carbon preference index (CPI) and average chain length (ACL) of *n*-alkanes in plants [9,35]. In this study, when compared with those in 2018, the PI of ACL_alkanes_ decreased whereas CPI_alkanes_ increased in both Hongyuan and Qilian. Such changes indicated that transplanted plants adapted to the new environment by altering both wax deposition as well as wax compound chain length distributions. 

### 3.2. The Phenotypic Plasticity of Waxes Traits Improved the Plant Adaptations

Overall, *P. viviparum* and *L. nanum* had better leaf trait phenotype in Hongyuan than in Qilian. Although the PI of plant height and leaf morphological traits were close to 1, their measured values reduced significantly for the six tested plant species in new environment, suggesting that some plant functional traits could not adjust towards better phenotype. Therefore, such plants may not be able to express the best phenotype in a given environment. Climate change alters the availability of resources and the conditions that are crucial to plant performance, and, therefore, plants have to balance between survival and performance in new environments [39]. Studies have shown that plasticity in itself is disadvantageous, unless it contributes to increased fitness and adaptability under climate change [18]. For example, warmer climate caused plasticity in *Eriophorum vaginatum*, which inhibited plant nitrogen metabolism, photosynthesis and growth [40]. Populations originating from warmer and more variable climates showed higher phenotypic plasticity. Kreyling et al. [41] reported that phenotypic plasticity can itself be considered as a trait subject to local adaptation to climate for a common grass. A study using Arabidopsis accessions also indicated that plants originating from warmer climates having a higher leaf dry matter content [42]. 

Previous researches demonstrated that the evolution of phenotypic plasticity was an important factor for population persistence in a variety of natural systems, however, whether the reason of increased plasticity is the result of climate change or an emergent trait from selection at shorter scale is still not clear [43,44]. In our study, the leaf wax traits of transplanted plants were more and more closer with those of local plants from 2018 to 2019, suggesting that plant species could adjust wax traits plasticity in order to survival in a new environment in a short time. As perennial herbs, such changes of wax traits plasticity might improve the plant adaptations not only in one growing season, but also in the following growing seasons. However, a study using a temperate tree species has shown that adaptive evolution in response to climate change might be limited by a lack of heritable variation [45]. Though cuticular waxes are very sensitive to changing environment and mainly show phenotypical plasticity under various environments, the responses of the chain length distribution of alkanes and alcohols implied that such changes might be heritable. A study using perennial *Leymus chinensis* has also shown that wax deposition patterns of plant populations formed during adaptations to their long-term growing environments could inherit in their progenies and exhibit such inheritance even these progenies were exported to new environments [46].

In response to different biotic and abiotic conditions, plant leaves will form various types and structures in order to grow and reproduce efficiently [47]. For example, leaf area has closest correlation with biomass, and leaf thickness is related to the environmental change [48]. In this study, the leaf length, width and thickness were positively correlated in Hongyuan, the leaf width and leaf thickness were positively correlated in Qilian, and the leaf wax characteristics showed low correlation with other leaf functional traits (Appendix A) and were generally separated in PCA analysis. These results suggested that leaf cuticular waxes might have its own unique function by changing its structure and compositions to changing environments. Plant adaptive responses are key mechanisms to cope with a changing environment [49]. Both leaf traits and cuticular waxes were involved in their contributions in improving plant adaptations in new environments. However, the phenotypic plasticity of wax traits was lower than chlorophyll and other plant morphological trait (especially that of height) both in Hongyuan and Qilian. This indicated that our selected species adopted a strategy to primarily change their morphology in changing environments. However, SEM analysis further indicated that leaf traits directly affected the wax coverage and wax composition content, particularly when environmental resources changed from “bad” (Qilian) to “better” (Hongyuan) (Figure 6). This implied that co-variation might exist between leaf traits and leaf cuticular waxes, depending upon environmental pressure. Oliveras et al. [50] also found that trait co-variation was strongly dependent on the local environment, and thus global trait co-variation relationships might not always apply at smaller scales. 

In conclusion, our results showed that the alpine meadow herbs would adjust both wax deposition and chain length distributions of alkanes to adapt to the changing environments. As perennial herbs, some of the changes of cuticular waxes might be inherited and help the plants to adapt to similar stress in the coming years, and therefore, such responses might be important for the alpine plants to deal with the climate changes. The total wax coverage and wax compositions also differed between the local and transplanted plants, suggesting that both environmental and genetic factors controlling the wax depositions on alpine herbs. However, the six tested plant species differed in their phenotypic plasticity and co-variations were observed between leaf functional traits and leaf cuticular waxes, depending upon the environmental pressures. Future studies are needed to clarify the relationships between the phenotypic plasticity of leaf cuticular waxes and the responses of plant communities as well as the ecological functions of alpine meadow. 

## 4. Materials and Methods

### 4.1. Experimental Design

This study was conducted outdoors at two experimental sites in eastern edge of Qinghai-Tibetan Plateau. One was at Hongyuan (Altitude 3434 m, N 32°49′56.40″, E 102°34′59.24″) and another was at Qilian (Altitude 3701 m, 37°58′48.68″, E 100°13′41.94″) (Figure 6). The average annual temperature and annual precipitation were 2.97 °C and 911 mm at Hongyuan, and 2.52 °C and 472 mm at Qilian (Appendix A). The steppe type of the two sites belonged to alpine meadow, which was mainly used for sheep and yak grazing. No other managements (irrigation and fertilization) were applied. Based on previous plant investigation, six plant species that were distributed at both experimental sites were selected, including *Plantago asiatica*, *Polygonum viviparum*, *Anaphalis lactea*, *Kobresia humilis*, *Leontopodium nanum* and *Potentilla chinensis*. In June 2017, when the plants at both sites were in their early vegetative stages, the plants were manually dug out (0–30 cm) using shovel at both sites. The plants at Hongyuan were transplanted to Qilian, whereas the plants at Qilian were transplanted to Hongyuan. The local plants were regarded as control. At both sites, ca. 100 plants for each species were transplanted in soils free of local plants (plot size: 5 m × 5 m), watered, and fenced (no grazing). The number of *K. humilis* reached 600. The plots were weeded every 20 days to keep the local plants away from the transplanted plots. No other management were applied. The soil was a typical meadow soil at both sites, with soil basic properties varied greatly between two sites. The soil pH level was 6.21 at Hongyuan and 8.48 at Qilian (see detailed properties from Appendix A). No sampling was done in 2017. 

### 4.2. Soil Chemical Analysis

Before reciprocal transplanting, soil samples were collected from both sites in June of 2017. In total, nine soil cores (6 cm in diameter) from three quadrats were collected at 0–20 cm soil layers, bulked into one composite soil sample. Organic matter was determined by oxidation with potassium dichromate in a concentrated sulfuric acid medium and the excess dichromate was measured using Mohr’s salt (K_2_Cr_2_O_7_-H_2_SO_4_) [51]. Dried soils (1.000 g) were digested in 5 mL H_2_SO_4_ and then determined for soil total nitrogen by Kjeldahl method [52]. Total potassium was measured using NaOH fusion and flame photometry [52]. Dried soils (2.500 g) were analyzed for available phosphorus using Mo-Sb colorimetric procedure [52]. Dried soils (2.000 g) were digested in 10 mL 1 mol L^−1^ NaOH solution for 24 h and titrated with 0.01 mol L^−1^ 1/2 H_2_SO_4_ for alkali dispelled nitrogen [52]. Dried soils (5.000 g) were analyzed for available potassium using flame photometer method [52]. Soil pH value was determined in a soil:water solution (1:5) using a pH meter [52].

### 4.3. Plant Traits Measurements

One year after transplanting, the plant traits were measured in July of 2018 and 2019, when most plants were in their blossoming stage. Shoot height was measured from the top of the soil to the tallest shoot tip including the apical leaf, and node number per stem of the five tallest and healthy shoots of each plant species. Ten fully expanded, mature, and healthy leaves were chosen from the same leaf position of ten individuals of each plant species and were cut off with scissors. Leaf length (LL, cm) and width (LW, cm) were measured with rulers, and the leaf area (cm^2^ per individual) was measured using a scanner (Cano Scan LIDE 110, Japan) and Photoshop CS (Adobe, United States). The leaf thickness (LT) near the apex, near the base, and half-way between the two points were measured by using a Mitutoyo 547–500S Digimatic Digital Thickness Gage. All leaves were then placed in a drying oven for a minimum of 48 h at 65 °C, and the final dry mass (g) was measured with an electronic balance. SLA (cm^2^ g^−1^) was calculated as: SLA = leaf area/leaf dry weight. The content of chlorophyll was measured using SPAD. 

### 4.4. Leaf Cuticular Waxes Extraction

In July of 2018 and 2019, approximately 30 leaves (one leaf per individual) at same leaf position were randomly sampled from healthy plants for each plant species, placed in absorbent paper, and dried (the absorbent papers were changed every other day during the first 7 days, then every 3 days until the plants were dried). Before wax extraction, the leaves were photographed, and the leaf areas were measured by pixel counting the photo using ImageJ software. For each plant species, five leaves were grouped into one replicate and in total three replicates at each site. For *K. humilis*, 10 leaves for each replicate. Then, the dried leaves were extracted in 50 mL chloroform containing 10 μg tetracosaneas as internal standard at room temperature for 1 min [7]. The extracts were dried using N_2_ at 40 °C, and derivatised using 30 μL pyridine and 30 μL BSTFA (bis-(trimethylsilyl)trifluoroacetamide) for 45 min at 70 °C. The surplus BSTFA was evaporated under N_2_ and the sample was re-dissolved in 500 μL chloroform for GC analysis. 

### 4.5. GC and GC/MS Analysis

The GC analysis was carried out with 9790II gas chromatograph (Zhejiang Fuli Analytic Instruments Co., China). The GC column was DM-5 (30 m × 0.32 mm × 0.25 μm capillary column) (Dikma Technologies Inc., USA). N_2_ was served as carrier gas. The injector and flame ionization detector (FID) temperatures were set at 300 and 320 °C, respectively. The original temperature of the column started from 80 °C, increased to 260 °C by 15 °C min^−1^, remained 10 min, then increasedto 290 °C by 2 °C min^−1^, and further increased to 320 °C by 5 °C min^−1^, remained 10 min. The samples were further detected with a GCMS-QP2010 Ultra Mass Spectrometric Detector (Shimadzu Corp., Kyoto, Japan) for compound identification. He was the carrier gas and the column was HP-5 MS capillary column (30 m × 0.32 mm × 0.25 µm). Compounds were identified by comparing their mass spectra with published data and authentic standards. Amounts of leaf cuticular waxes per unit leaf area were calculated by the internal standard and were expressed as μg cm^−2^.

### 4.6. Statistical Analysis

Four-Way ANOVA analysis was performed to evaluate the effects of transplanting, plant species, sites and year on leaf parameters including plant height, leaf length, width, thickness, specific leaf area, total wax coverage, and the contents of wax compositions, using SPSS V17.0 (SPSS, IBM, Armonk, NY, USA). The principal component analysis and Spearman correlation analysis were performed in SPSS V17.0 (SPSS, IBM, Armonk, NY, USA). Structural equation modeling (SEM) was used to determine how pathways of trait factors affected phenotypic plasticity. The structural equation model analyses were undertaken using the “plspm” package in R version 3.6.0 (https://www.r-project.org/, R Core Team 2019, accessed on 24 September 2021) [53].

We refer to the method of Poynter and Eglinton [54] to calculate the average chain length (ACL) of alkanes, alcohols and the method of Mazurek and Simoneit [55] to calculate the carbon preference index (CPI) of alkanes.
(1)ACL =∑Cn×n/∑Cn
(2)CPI_alkanes=1⁄2((∑oddCn(26−35)/∑evenCn(26−36))+(∑oddCn(27−37)/∑evenCn(26−36)))

In the formula, Cn was the relative content of alkanes; Odd, Odd chain length compound; even, even chain length compound. *n* was the number of carbon atoms of *n*-alkanes, *n*-alcohols and *n*-acids; the value range of *n* was 20 ≤ *n* ≤ 37.

The trait response coefficient (RC) of different index in two sites, was calculated according to following equation [56]: 

Different sites:(3)$${\mathrm{RC}}_{\mathrm{Hongyuan}}={\mathrm{Mean}}_{\mathrm{HT}}/{\mathrm{Mean}}_{\mathrm{QL}}$$
(4)$${\mathrm{RC}}_{\mathrm{Qilian}}={\mathrm{Mean}}_{\mathrm{QT}}/{\mathrm{Mean}}_{\mathrm{HL}}$$

Same site: (5)$${\mathrm{RC}}_{\mathrm{Hongyuan}}={\mathrm{Mean}}_{\mathrm{HT}}/{\mathrm{Mean}}_{\mathrm{HL}}$$
(6)$${\mathrm{RC}}_{\mathrm{Qilian}}={\mathrm{Mean}}_{\mathrm{QT}}/{\mathrm{Mean}}_{\mathrm{QL}}$$

In the formula, HT was the plants transplanted from Qilian to Hongyuan; QL is the plants local in Qilian; QT is the plants transplanted from Hongyuan to Qilian; HL is the plants local in Hongyuan; Mean referred to the average of certain index.

Phenotypic Plasticity Index (PI) was calculated according the method described by Valladares et al. [57]. PI = |(mean value for one site-mean value for the other site)|/maximum mean value between two sites. The mean value of each trait was the average of three replicates. The PI ranged from 0 to 1, with 0 represented no plasticity and 1 represented the highest level of plasticity possible.

We performed Horn’s parallel analysis to adjust the number of components for principal component analysis (PCA) on all measured traits in 2019, aiming to visualize the overall differences in trait intra-variations at different environmental sites. Furthermore, we estimated the trait-trait interdependence within each environment by Spearman correlation coefficient among pairwise measured traits.

## Figures and Tables

**Figure 1 plants-11-00120-f001:**
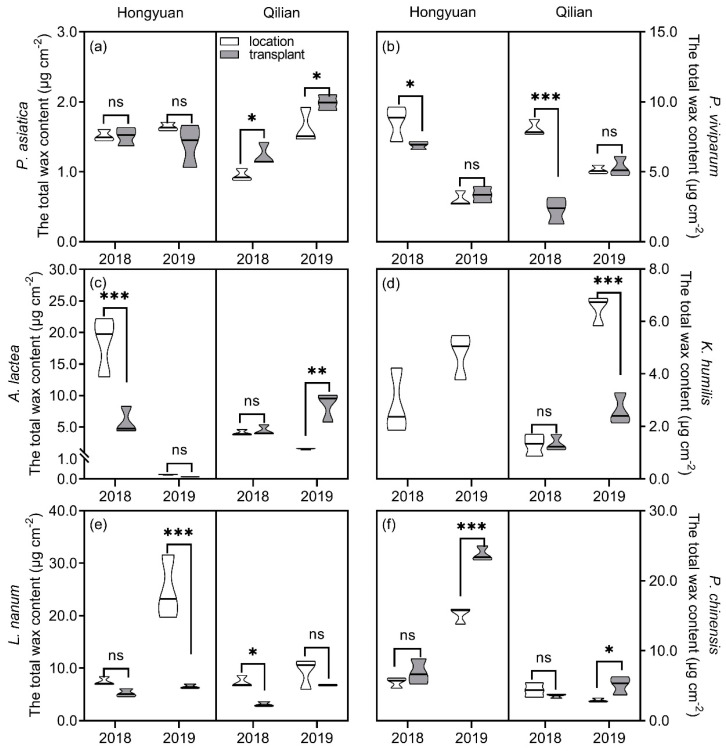
The total wax coverage on leaves of six perennial alpine herbs at Hongyuan and Qilian in 2018 and 2019. (**a**–**f**) represented *Plantago asiatica*, *Polygonum viviparum*, *Anaphalis lactea*, *Kobresia humilis*, *Leontopodium nanum*, *and Potentilla chinensis*, respectively. Each violin area represented the distribution of samples within three replicates and the black line represented median. *, significant differences between local and transplanted plants according to the least significant different test. *, 0.01 < *p* < 0.05; **, 0.001 < *p* <0.01; ***, *p* < 0.001; ns, no significantly difference.

**Figure 2 plants-11-00120-f002:**
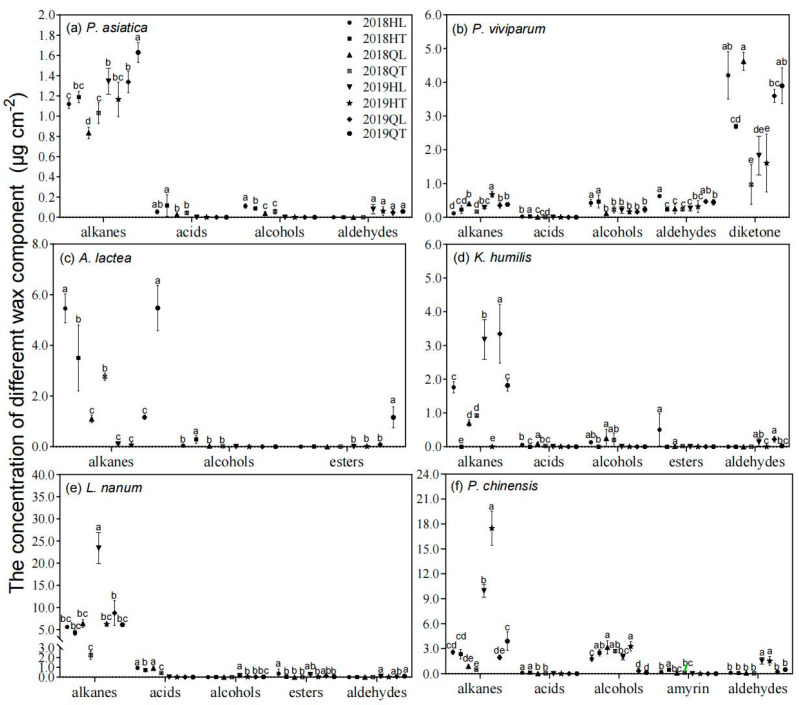
The influence of transplanting on the contents of leaf wax compositions in 2018 and 2019. HL, local plants at Hongyuan; HT, the transplanted plants from Qilian to Hongyuan; QL, local plants at Qilian; QT, the transplanted plants from Hongyuan to Qilian. Scatter plots represented the means of wax composition contents (alkane, alcohol, alkanoic acid, aldehyde and esters). The error bars indicated standard deviation of means (SD). Different lowercase letters showed significant differences between local and transplanted plants within each wax composition according to the least significant different test (*p* < 0.05).

**Figure 3 plants-11-00120-f003:**
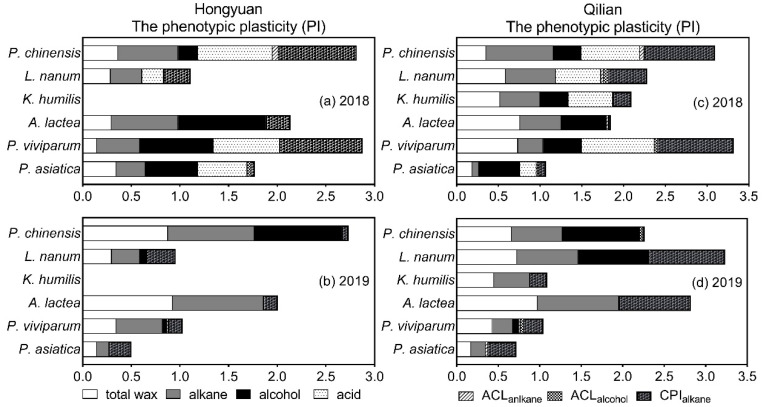
Phenotypic plasticity indices (PI) of leaf wax traits at Hongyuan and Qilian in 2018 and 2019. (**a**), PI of leaf wax traits at Hongyuan in 2018; (**b**), PI of leaf wax traits at Hongyuan in 2019; (**c**), PI of leaf wax traits at Qilian in 2018; (**d**), PI of leaf wax traits at Qilian in 2019. ACL_alkane_, average chain length of alkane; ACL_alcohol_, average chain length of alcohol; CPI_alkane_, carbon preference index of alkane. Plasticity index ranged from 0 (no plasticity) to 1 (maximal plasticity) according to the method described by Valladares et al. (2000).

**Figure 4 plants-11-00120-f004:**
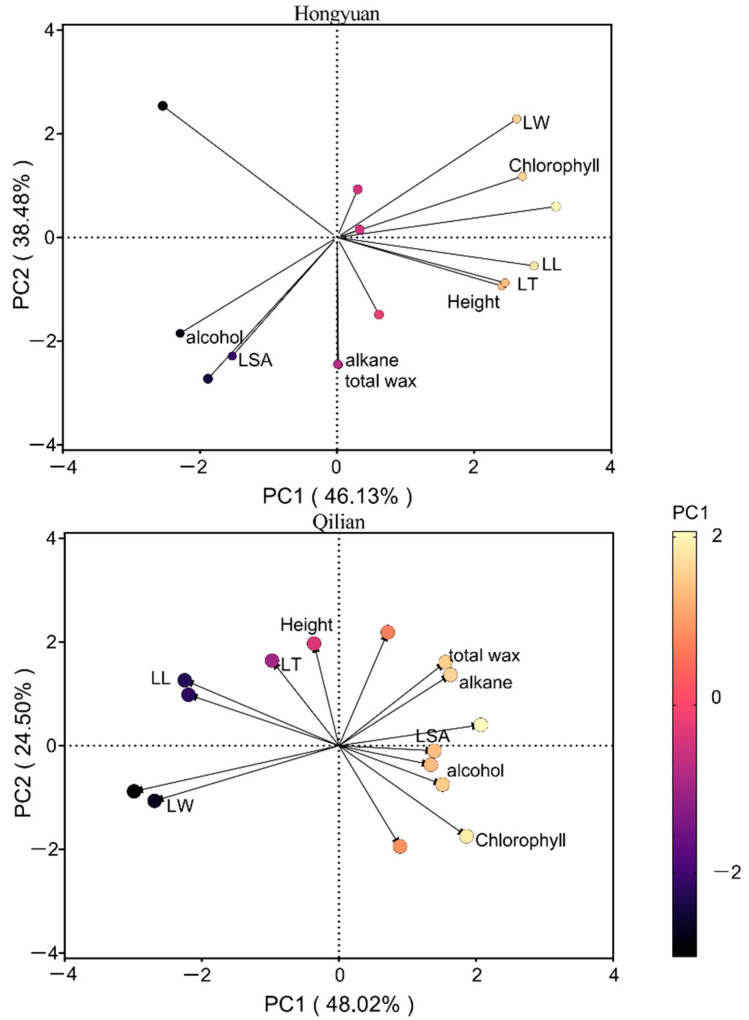
Principal components analysis of the leaf functional traits and leaf cuticular waxes traits at Hongyuan and Qilian in 2019. The colour gradient of the circles represented the mean contribution (percentage) to the PC1 of a given variable. LL, leaf length; LW, leaf width; LT, leaf thickness; LSA, leaf specific area.

**Figure 5 plants-11-00120-f005:**
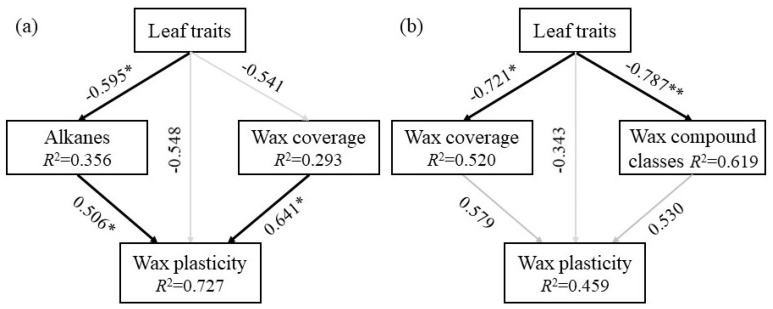
Structural equation model with indirect and direct effects of leaf functional traits and cuticular waxes traits on wax plasticity of perennial alpine herbs in 2018 (**a**) and 2019 (**b**). The Goodness of Fit was 0.4418 for Hongyuan and 0.4892 for Qilian. *, 0.01< *p* < 0.05; **, 0.001< *p* < 0.01.

**Figure 6 plants-11-00120-f006:**
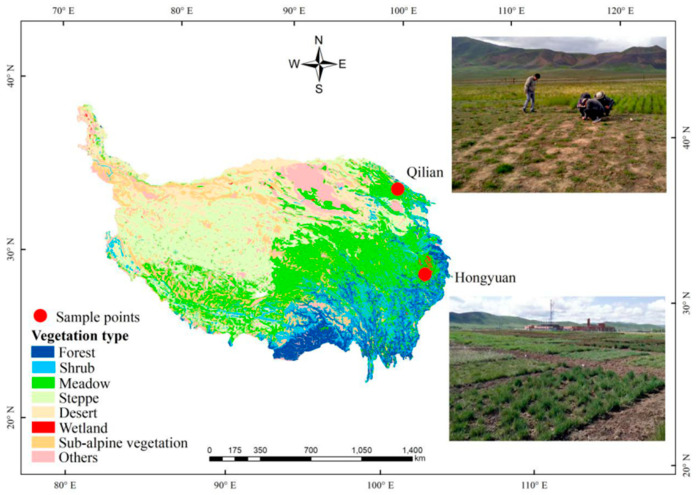
Geographical locations of the two experimental sites (Hongyuan and Qilian) in this study.

**Table 1 plants-11-00120-t001:** Analysis of variance of main effects (year, plant species, site and transplanting) and their interactions on morphological parameters, chlorophyll content, total wax, wax components, ACL_alkane_, ACL_alcohol_ and CPI_alkane_.

	Year	Plant Species	Site	Transplanting (T)	Y × P	Y × S	Y × T	P × S	P × T	S × T	Y × P × S	Y × P × T	P × S × T
Height	3.676 ns	16.310 **	219.332 ***	61.054 ***	5.243 *	10.334 *	0.044 ns	16.703 ***	1.313 ns	11.001 *	5.509 *	0.594 ns	3.741 ns
LL	5.286 ns	22.351 **	33.243 **	8.588 *	1.487 ns	3.273 ns	2.651 ns	5.793 *	3.007 ns	0.174 ns	1.724 ns	1.239 ns	6.459 *
LW	16.849 **	188.048 ***	26.282 **	14.684 **	3.436 ns	0.173 ns	3.815 ns	12.153 **	2.095 ns	5.212 ns	0.211 ns	0.958 ns	3.911 ns
LT	2.053 ns	30.140 ***	67.430 ***	3.967 ns	1.088 ns	1.027 ns	1.496 ns	14.656 **	12.937 **	0.011 ns	0.901 ns	1.000 ns	10.264 **
LSA	8.912 *	13.274 **	0.000 ns	7.548 *	0.670 ns	3.025 ns	1.730 ns	2.972 ns	1.500 ns	0.537 ns	4.862 *	1.538 ns	4.521 *
chlorophyll	1.547 ns	3805.702 ***	1434.214 ***	876.592 ***	1.247 ns	0.346 ns	1.272 ns	371.77 ***	316.469 ***	1497.712 ***	1.145 ns	1.167 ns	314.742 ***
total wax	2.548 ns	8.112 *	6.898 *	4.348 ns	5.376 *	0.032 ns	0.479 ns	3.452 ns	2.531 ns	1.242 ns	4.838 *	2.285 ns	1.750 ns
alkanes	8.729 *	9.329 **	4.330 ns	0.966 ns	3.519 ns	0.913 ns	0.018 ns	2.353 ns	2.578 ns	0.902 ns	2.148 ns	1.221 ns	0.705 ns
alcohols	184.844 ***	1236.322 ***	82.709 ***	11.553 *	91.938 ***	179.882 ***	1.215 ns	53.043 ***	9.968 **	32.439 **	221.710 ***	4.592 *	37.705 ***
acids	188.534 ***	111.142 ***	3.139 ns	8.719 *	111.142 ***	3.139 ns	8.719 *	1.331 ns	6.741 *	1.068 ns	1.331 ns	6.741 *	0.986 ns
ACL_alkane_	3.161 ns	1107.626 ***	669.139 ***	879.018 ***	5.795 *	1.351 ns	0.086 ns	852.105 ***	818.575 ***	824.850 ***	4.497 *	1.545 ns	941.012 ***
CPI_alkane_	10.078 *	11.481 **	17.429 **	0.816 ns	11.619 **	12.962 *	0.015 ns	8.916 **	0.340 ns	2.409 ns	10.078 **	0.244 ns	0.633 ns
ACL_alcohol_	111.223 ***	30.430 ***	0.613 ns	1.225 ns	22.662 **	0.889 ns	0.984 ns	1.079 ns	0.964 ns	0.896 ns	1.013 ns	1.013 ns	1.041 ns
ACL_acid_	326.320 ***	15.474 **	0.387 ns	1.290 ns	15.474 **	0.387 ns	1.290 ns	0.994 ns	0.855 ns	0.570 ns	0.994 ns	0.855 ns	1.086 ns

Abbreviations: Y, year; P, plant species; S, site; T, transplanting; LL, leaf length; LW, leaf width; LT, leaf thickness; LSA, leaf specific area; ACL_ankane_, average chain length of alkane; ACL_alcohol_, average chain length of alcohol; ACL_acid_, average chain length of acid; CPI_ankane_, carbon preference index of alkane. *, 0.01 < *p* < 0.05; **, 0.001 < *p* < 0.01; ***, *p* < 0.001; ns, no significantly difference.

**Table 2 plants-11-00120-t002:** The response coefficient of the leaf cuticular wax traits of the six alpine herbs at Hongyuan and Qilian in 2018 and 2019. HT was the plants transplanted from Qilian to Hongyuan; QL is the plants local in Qilian; QT is the plants transplanted from Hongyuan to Qilian; HL is the plants local in Hongyuan; Mean referred to the average of certain index.

		2018				2019			
Plant Speices	Indexes	HT/HL	HT/QL	QT/QL	QT/HL	HT/HL	HT/QL	QT/QL	QT/HL
*P. asiatica*	alkanes	1.061	1.423	1.236	0.922	0.867	0.872	1.219	1.212
	acids	2.160	4.285	1.595	0.804	-	-	-	-
	alcohols	0.776	2.185	1.419	0.504	-	-	-	-
	aldehydes	-	-	-	-	0.769	1.274	1.198	0.722
	total wax	0.997	1.594	1.302	0.815	0.863	0.855	1.218	1.229
*P. viviparum*	alkanes	1.929	0.560	0.416	1.434	2.355	1.914	1.072	1.319
	acids	1.352	3.163	0.304	0.130	-	-	-	-
	alcohols	1.098	4.115	2.020	0.539	0.771	1.044	1.469	1.084
	aldehydes	0.364	0.899	0.938	0.380	1.201	0.679	0.952	1.683
	diketone	0.640	0.583	0.210	0.230	0.880	0.447	1.084	2.132
	total wax	0.807	0.853	0.280	0.265	1.078	0.620	1.045	1.817
*A. lactea*	alkanes	0.642	3.189	2.515	0.506	0.725	0.065	4.710	52.847
	alcohols	6.904	9.718	0.641	0.456	-	-	-	-
	esters	-	-	-	-	1.653	0.106	13.456	209.515
	total wax	0.314	1.391	1.032	0.233	0.741	0.070	5.483	57.968
*K. humilis*	alkanes	0.000	0.000	1.320	0.525	0.000	0.000	0.542	0.571
	acids	0.000	0.000	0.294	0.466	-	-	-	-
	alcohols	0.000	0.000	0.810	1.521	-	-	-	-
	esters	0.000	0.000	1.356	0.038	-	-	-	-
	aldehydes	-	-	-	-	0.000	0.000	0.109	0.173
	total wax	0.000	0.000	1.033	0.478	0.000	0.000	0.454	0.564
*L. nanum*	alkanes	0.772	0.677	0.349	0.398	0.266	0.713	0.698	0.261
	acids	0.767	0.778	0.465	0.458	-	-	-	-
	alcohols	-	-	-	-	0.225	1.080	0.695	0.145
	esters	0.125	-	-	0.000	0.143	0.313	0.325	0.148
	aldehydes	-	-	-	-	0.227	0.375	1.849	1.119
	total wax	0.702	0.712	0.422	0.416	0.262	0.700	0.729	0.273
*P. chinensis*	alkanes	0.913	2.612	0.552	0.193	1.758	8.958	1.992	0.391
	acids	0.969	4.182	1.291	0.299	-	-	-	-
	alcohols	1.393	0.794	0.858	1.504	1.635	9.721	0.402	0.068
	amyrin	2.365	5.171	1.323	0.605	-	-	-	-
	aldehydes	0.686	0.802	0.410	0.351	0.990	6.774	2.186	0.320
	total wax	1.260	1.577	0.810	0.647	1.616	8.281	1.764	0.344

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
