# Peer review of "Phenotypic Plasticity and Local Adaptation of Leaf Cuticular Waxes Favor Perennial Alpine Herbs under Climate Change"

_plants, 2021, doi:10.3390/plants11010120_

Round 1

Reviewer 1 Report

This work needs to be edited for clarity and importance.  

Include elevations in the abstract and be explicit about transplanting between a drier and wetter meadow

Need more evidence for why cuticular wax matters - will it change herbivory rates for animals? Will more wax mean fewer flowers, thus altering the population size?  Provide something more than plants change with climate change - that is already well established.

Introduction needs clearly stated hypotheses, not a broad objective

Table 1 is not useful

Author Response

Comments and Suggestions for Authors

  1. This work needs to be edited for clarity and importance.

Response: we have carefully considered the suggestion and make some changes.

  1. Include elevations in the abstract and be explicit about transplanting between a drier and wetter meadow

Response: Elevations were added as suggested. Compared with annual temperature, precipitation difference was obvious between two sites. However, other factors such ultraviolet radiations, wind speed, etc, might also differ between the two sites, which would influence wax deposition. Wax deposition might be a comprehensive response to various environmental factors.

  1. Need more evidence for why cuticular wax matters - will it change herbivory rates for animals? Will more wax mean fewer flowers, thus altering the population size?  Provide something more than plants change with climate change - that is already well established.

Response: Good comments. As an important secondary metabolite, the change of leaf cuticular wax may have various ecological influences including herbivory rates for animals. We added references reporting the influences of wax on digestibility, insect activity and attachment and effectiveness of predatory insects. However, no reference is available as for their influence on flowers and thus population size.

  1. Introduction needs clearly stated hypotheses, not a broad objective

Response: As Reviewer suggested, a hypotheses was added.

  1. Table 1 is not useful

Response: We kept Table 1. It presented the direct and interactive effects years, plant species, experimental sites and transplanting on leaf traits of alpine meadow plants.

Reviewer 2 Report

The concerned manuscript is interesting, and can be published

Author Response

The concerned manuscript is interesting, and can be published

Response: Thanks for the comments.

Reviewer 3 Report

In this paper by Yao et al., six perennial herbs were reciprocally transplanted at two sites of Qinghai-Tibetan Plateau, Hongyuan and Qilian. Leaf cuticular waxes traits and leaf functional traits were measured in 2018 and 2019, respectively. The authors explore the phenotypic plasticity of cuticular waxes as well as leaf functional traits in alpine meadow plants under climate change. The results showed that the environments and genetic factors affected the wax deposition. Moreover, the covariations existed between leaf cuticular waxes and leaf functional traits. They demonstrated that alpine herbs regulate both wax deposition and chain length distribution to adapt to changing environments, showing climatic adaptation. This manuscript is well written. In my opinion, only a few issues should be strengthened before publication.

Major concerns:

  1. In the “Results” section, many descriptions of results do not add corresponding Figures or Tables.
  1. The abbreviations in Table 1 (ACLalkane, CPIalkane, ACLalcohol, ACLacid) are inconsistent with the notes (anACL, olACL, anCPI).
  1. In the Supplementary Figure 2 and Figure 3, it is not clear what the abbreviations (LL, LW, LT, LSA, ACLalkane, ACLalcohol, CPIalkane) stand for.
  1. The section of “Discussion” is not very clear. I suggest that it can be divided into several sections and add subtitles to highlight the main findings.

Minor concerns:

  1. Line 52: “had obvious long chain alkane predominance” should be “had obvious predominance of long-chain alkane”.
  2. Line 53: “plant” should be “plants”.
  3. Line 63: “trait” should be “traits”.
  4. Line 64, 72: “environment” should be “environments”.
  5. Line 69, 224, 229, 240, 330, 410: “wax” should be “waxes”.
  6. Line 69: “benefit our understanding” should be “benefit to our understanding”.
  7. Line 77: “with” should be “of”.
  8. Line 137: “preference index” should be “preference index of alkane”.
  9. Line 152: “phenotypic plasticity (PI)” should be “phenotypic plasticity index (PI)”.
  10. Line 216: “The SEM” should be “The structural equation model (SEM)”.
  11. Line 249: “Additional” should be “Additionally”.
  12. Line 262: “plant species” should be “plant species (Fig. 2)”.
  13. Line 273: “demonstrate” should be “demonstrated”.
  14. Line 339: “from “bad” (Qilian) to “better" (Hongyuan)” should be “from “bad” (Qilian) to “better" (Hongyuan) (Fig. 6)”.
  15. Line 363: “2.52” should be “2.52°C”.

Author Response

Comments and Suggestions for Authors

In this paper by Yao et al., six perennial herbs were reciprocally transplanted at two sites of Qinghai-Tibetan Plateau, Hongyuan and Qilian. Leaf cuticular waxes traits and leaf functional traits were measured in 2018 and 2019, respectively. The authors explore the phenotypic plasticity of cuticular waxes as well as leaf functional traits in alpine meadow plants under climate change. The results showed that the environments and genetic factors affected the wax deposition. Moreover, the covariations existed between leaf cuticular waxes and leaf functional traits. They demonstrated that alpine herbs regulate both wax deposition and chain length distribution to adapt to changing environments, showing climatic adaptation. This manuscript is well written. In my opinion, only a few issues should be strengthened before publication.

Major concerns:

  1. In the “Results” section, many descriptions of results do not add corresponding Figures or Tables.

Response: Checked and added when it is necessary.

  1. The abbreviations in Table 1 (ACLalkane, CPIalkane, ACLalcohol, ACLacid) are inconsistent with the notes (anACL, olACL, anCPI).

Response: Checked and revised.

  1. In the Supplementary Figure 2 and Figure 3, it is not clear what the abbreviations (LL, LW, LT, LSA, ACLalkane, ACLalcohol, CPIalkane) stand for.

Response: Added. LL, leaf length; LW, leaf width; LT, leaf thickness; LSA, leaf specific area; ACLankane, average chain length of alkane; ACLalcohol, average chain length of alcohol; CPIalkane, carbon preference index.

  1. The section of “Discussion” is not very clear. I suggest that it can be divided into several sections and add subtitles to highlight the main findings.

Response: Subtitles were added as suggested.

Minor concerns:

  1. Line 52: “had obvious long chain alkane predominance” should be “had obvious predominance of long-chain alkane”.

Response: Revised as suggested.

  1. Line 53: “plant” should be “plants”.

Response: Revised as suggested.

  1. Line 63: “trait” should be “traits”.

Response: Revised as suggested.

  1. Line 64, 72: “environment” should be “environments”.

Response: Revised as suggested.

  1. Line 69, 224, 229, 240, 330, 410: “wax” should be “waxes”.

Response: Revised as suggested.

  1. Line 69: “benefit our understanding” should be “benefit to our understanding”.

Response: Revised as suggested.

  1. Line 77: “with” should be “of”.

Response: Revised as suggested.

  1. Line 137: “preference index” should be “preference index of alkane”.

Response: Revised as suggested.

  1. Line 152: “phenotypic plasticity (PI)” should be “phenotypic plasticity index (PI)”.

Response: Revised as suggested.

  1. Line 216: “The SEM” should be “The structural equation model (SEM)”.

Response: Revised as suggested.

  1. Line 249: “Additional” should be “Additionally”.

Response: Revised as suggested.

  1. Line 262: “plant species” should be “plant species (Fig. 2)”.

Response: Revised as suggested.

  1. Line 273: “demonstrate” should be “demonstrated”.

Response: Revised as suggested.

  1. Line 339: “from “bad” (Qilian) to “better" (Hongyuan)” should be “from “bad” (Qilian) to “better" (Hongyuan) (Fig. 6)”.

Response: Revised as suggested.

  1. Line 363: “2.52” should be “2.52°C”.

Response: Revised as suggested.

Reviewer 4 Report

Yao and co-authors investigated patterns of phenotypic plasticity in cuticular waxes and leaf functional traits in alpine plants. The study was based on reciprocal transplanting of six vascular species at two alpine-meadow sites in Qinghai-Tibetan Plateau (China). The topic is of potential interest to plant ecologists because adaptation of cuticular waxes to changing environmental conditions may represent an important tool for alpine plants to cope with environmental stress caused by climate change.

I have a big concern with the rationale of the study as regards both the choice of the response variables and, especially, of the study sites. As far as I understand, although the two sites are quite distant from each other (almost 6 degrees in latitude) and about 300 m apart in elevation, mean annual temperature is much the same in the two sites (2.97 °C vs. 2.52 °C). Therefore, the only true climatic difference between them consists in precipitation being almost double in Hongyuan (911 mm) compared to Qilian (472 mm). So, a reader imagines that the authors wished to test how the response variables examined will vary in response to different water availability in the two sites but there is no explicit statement in this sense. Indeed, the authors only say that the environmental resources changed from "better" (Hongyuan) to "bad" (Qilian) as a consequence of transplanting, see l. 217-218.

In addition, and probably most important, a reader does not know in which direction the variables examined are expected to vary because of transplanting. Indeed, the authors did not formulate any hypothesis to this regard as they only state (l. 99-100) that the objective of their study was to elucidate the contributions of leaf cuticular waxes to alpine plant adaptations under climate change.

Based on the reasoning conveyed in the Introduction section, one may expect that the plants transplanted to the dry site have higher total was content (l. 43-44) and longer-chain hydrocarbons (l. 49-50). No straightforward directions of variation was described for leaf functional traits (l.57-70) which makes it extremely difficult for a reader to understand how transplanting influenced leaf functional traits.

In the absence of such essential information it is impossible to evaluate how the results of this study really add to our understanding of the ecology of mountain meadows.

Author Response

Comments and Suggestions for Authors

Yao and co-authors investigated patterns of phenotypic plasticity in cuticular waxes and leaf functional traits in alpine plants. The study was based on reciprocal transplanting of six vascular species at two alpine-meadow sites in Qinghai-Tibetan Plateau (China). The topic is of potential interest to plant ecologists because adaptation of cuticular waxes to changing environmental conditions may represent an important tool for alpine plants to cope with environmental stress caused by climate change.

  1. I have a big concern with the rationale of the study as regards both the choice of the response variables and, especially, of the study sites. As far as I understand, although the two sites are quite distant from each other (almost 6 degrees in latitude) and about 300 m apart in elevation, mean annual temperature is much the same in the two sites (2.97 °C vs. 2.52 °C). Therefore, the only true climatic difference between them consists in precipitation being almost double in Hongyuan (911 mm) compared to Qilian (472 mm). So, a reader imagines that the authors wished to test how the response variables examined will vary in response to different water availability in the two sites but there is no explicit statement in this sense. Indeed, the authors only say that the environmental resources changed from "better" (Hongyuan) to "bad" (Qilian) as a consequence of transplanting, see l. 217-218.

Response: We agree with the comments from reviewer. The two sites, though differed little in annual temperatures, the great difference in latitudes may result in differences of precipitations as observed in this study, ultraviolet radiation, and wind speed, which may influence the depositions of leaf cuticular waxes.

  1. In addition, and probably most important, a reader does not know in which direction the variables examined are expected to vary because of transplanting. Indeed, the authors did not formulate any hypothesis to this regard as they only state (l. 99-100) that the objective of their study was to elucidate the contributions of leaf cuticular waxes to alpine plant adaptations under climate change.

Response: We added a hypothesis in Introduction. However, it is still difficult to state in which direction the variables examined are expected to vary. As we discussed in Introduction, the depositions of leaf cuticular waxes would increase or decrease, depending on the complicated environmental conditions. For example, Dodd (2003) reported that greater weighted mean alkane chain length at low and at high elevations was possibly a result of adaptation to minimize cuticular permeability due to high summer temperatures at low elevation and freezing causing physiological drought at high elevations.

  1. Based on the reasoning conveyed in the Introduction section, one may expect that the plants transplanted to the dry site have higher total was content (l. 43-44) and longer-chain hydrocarbons (l. 49-50). No straightforward directions of variation was described for leaf functional traits (l.57-70) which makes it extremely difficult for a reader to understand how transplanting influenced leaf functional traits.

Response: We specified some examples of leaf functional traits. For example, narrow leaves are associated with higher mean annual precipitation in Protea. Light-demanding species in the moist forest had a high leaf area ratio, whereas, to reduce transpiration and water loss, light-demanding species in the dry forest had a low leaf area ratio.

  1. In the absence of such essential information it is impossible to evaluate how the results of this study really add to our understanding of the ecology of mountain meadows.

Response: With the above-mentioned revisions, we believe the results of this study will add our understanding of the ecology of mountain meadows.

Round 2

Reviewer 4 Report

Although the paper has a bit improved with respect to the interpretation of some trends in leaf functional traits, the authors did not succeed in tackling the major weak point of the paper. Since the paper principally addresses leaf cuticular waxes (as the title itself tells), a reader legitimately expects to know how wax contents is expected vary, even in contrasting directions, in response to evironment. That 'phenotipic plasticity of plant cuticular waxes of alpine meadow will benefit plant adaptability' (l. 107-108) is a quite vague sentence that does not allow one to understand if the study really hits the scientific target addressed.

Author Response

Although the paper has a bit improved with respect to the interpretation of some trends in leaf functional traits, the authors did not succeed in tackling the major weak point of the paper.

Since the paper principally addresses leaf cuticular waxes (as the title itself tells), a reader legitimately expects to know how wax contents is expected vary, even in contrasting directions, in response to environment.

Response: The relationship between leaf cuticular wax depositions and the environment is complicated, particularly for alpine plants. This is much different from the research work for cultivated plant species. As in our previous studies, drought and enhanced UVB would increase the leaf wax depositions. In alpine region, the plant species receive various environmental stimuli, differentially influencing the wax depositions. However, we observed that the changes of environments did alter the wax depositions. And such changes of leaf wax depositions in accordance with the environmental changes provided the readers about the importance of leaf cuticular wax for plant adaptations.

That 'phenotypic plasticity of plant cuticular waxes of alpine meadow will benefit plant adaptability' (l. 107-108) is a quite vague sentence that does not allow one to understand if the study really hits the scientific target addressed.

Response: One more citation was added to explain the inheritable alterations of cuticular wax depositions induced by long term environmental changes. The hypothesis was clear and addressed the importance of 'phenotypic plasticity of plant cuticular waxes in plant adaptability' for alpine meadow.

Round 3

Reviewer 4 Report

 I have carefully read the new version that is almost identical to the previous one, except a sentence trying to set up a very weak and poor working hypothesis.